# Select and Schedule: An Efficient Hierarchical Optimizer for Blocking Job Shop Scheduling Problem with Massive Jobs

## Abstract

The Blocking Job Shop Scheduling Problem (BJSP) is a widely studied variant of the classic Job Shop Scheduling Problem. In BJSP, the blocking constraint requires a job to remain on its current machine until the next machine is available. This constraint substantially increases problem complexity, which in turn limits most existing scheduling algorithms to small-scale instances. However, we observe that this blocking constraint also has merit: it naturally restricts the number of jobs processed concurrently, thereby reducing the number of candidate jobs that must be considered at almost any decision point. Building on this insight, we propose a novel hierarchical optimization framework. The higher layer employs a neural network to select a small subset of jobs from a large candidate pool, while the lower layer uses a solver to schedule the selected jobs. Compared with traditional approaches that directly schedule large sets of jobs, our method achieves significantly lower computational complexity and scales almost linearly with the number of jobs. This scalability enables us to efficiently handle larger instances that are previously intractable. Experimental results demonstrate that, on large-scale benchmarks and under comparable runtime budgets, our approach improves solution quality by an average of 11%, while continuing to deliver high-quality solutions within reasonable runtimes for even larger instances.

## 1 Introduction

Job-Shop Scheduling Problem (JSP) is a classical and widely studied combinatorial optimization problem with broad applications in manufacturing and automation (Kan, 2012; D'Ariano et al., 2007). Among numerous JSP variants (Li et al., 2022; Mascis & Pacciarelli, 2002), Blocking Job Shop Scheduling Problem (BJSP) is a realistic extension , which frequently encountered in domains such as chemical and pharmaceutical production, food processing, and automated warehousing (Hall & Sriskandarajah, 1996). In BJSP, a set of jobs must be processed on machines, where each job consists of a sequence of operations, each requiring a specific machine and a fixed processing time. Each machine can process only one job at a time. In contrast to classic JSP , BJSP models the more realistic scenario in which no intermediate buffer is available between machines. That is, once an operation completes, if its succeeding machine is not ready, the job remains on the current machine, blocking it from processing other jobs. This blocking constraint makes the problem significantly more complex. The goal is to determine the start times of all operations to minimize the completion time or makespan.

BJSP is further divided into two categories: Blocking No Swap (BNS) and Blocking With Swap (BWS) (Mascis & Pacciarelli, 2002). In BWS, jobs can be temporarily removed to allow others to move, while in BNS, such circular dependencies can cause deadlocks that must be avoided during scheduling. Figure 1 illustrates the specific differences. At time $t$ in BWS, Job 2 on Machine 1 is waiting to move to Machine 2, while Job 3 on Machine 2 is waiting to move to Machine 1. In BWS, it is allowed to simultaneously remove these two jobs from their machines and transfer them to the next machines, making the schedule feasible. However, in BNS, applying the same schedule immediately results in deadlock. Both variants have been proven to be NP-hard (Hall & Sriskandarajah, 1996). Our proposed method is applicable to both BWS and BNS, but the experiments primarily focus on the more challenging BNS case.

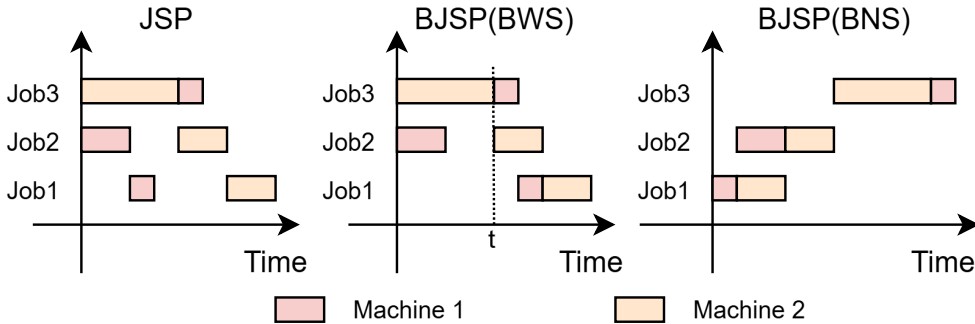

Figure 1: An example schedule for JSP, BNS and BWS. In JSP, Job 1 and Job 2 completed their first operation consecutively , but since Machine 1 is occupied, they wait in the buffer. In contrast, under BWS, when Job 2 finishes an operation, it remains on the current machine, preventing Job 1 from completing its first operation. In BNS, exchange in BWS at time $t$ is not allowed, and consequently its makespan becomes longer.

BJSP is highly challenging, as the blocking constraint drastically increases scheduling complexity. Existing research has mainly followed two directions: exact algorithms and metaheuristics (Dabah et al., 2017; Pranzo & Pacciarelli, 2016; Dabah et al., 2018; Rihane et al., 2022; Lange & Werner, 2019a; 2018). Exact methods can guarantee optimality but become infeasible beyond small instances due to their exponential time complexity (Dabah et al., 2016; 2018; Gmys et al., 2016). Within the class of metaheuristics, tabu search has demonstrated state-of-the-art performance (Lange & Werner, 2018; Mogali et al., 2021; Dabah et al., 2019; 2017). However, it typically requires a large number of iterations to reach high-quality solutions, resulting in relatively high computational cost and making it unsuitable for dynamic or real-time scheduling scenarios.

While the blocking constraint is commonly regarded as a source of computational difficulty, we observe that its inherent restriction on concurrency can be exploited to reduce computational complexity. Specifically, at any given time, the number of jobs that can be simultaneously in process—defined as jobs that have started their first operation but have not yet completed their final operation—is bounded by the number of machines. In high-quality schedules, the number of actively in-process jobs often approaches this upper bound. This implies that, until one of these jobs completes, other jobs typically need not be considered. In other words, in most situations, the scheduler can focus on a small subset of jobs rather than the entire job set.

Building on this insight, we design a hierarchical optimization framework ——*Select-and-Schedule (S&S)*. A high-level neural network dynamically selects a promising subset of jobs from the candidate pool, and a lower-level solver schedules them in detail. This S&S framework fundamentally differs from traditional approaches that attempt to schedule all jobs directly. By narrowing the effective problem size at each decision step, our method achieves computational complexity that grows nearly linearly with the number of jobs, making it possible to tackle large-scale BJSP instances that were previously intractable.

We validate our approach on challenging benchmarks and demonstrate that, under comparable runtime budgets, it improves solution quality by an average of 11% over state-of-the-art methods on large instances. Moreover, our framework continues to produce high-quality schedules within reasonable runtimes even as instance sizes grow further, highlighting its scalability and robustness. Our contributions can be summarized as follows:

- We identify a key structural property of BJSP: although the blocking constraint increases scheduling difficulty, it naturally limits concurrency. This observation allows the effective problem size at each decision point to be dramatically reduced, providing a new perspective for scalable scheduling.

- We propose a hierarchical optimization framework S&S, which integrates a high-level neural network to dynamically select promising job subsets and a lower-level solver to schedule

them.S&S narrows the effective problem size, achieving near-linear computational complexity growth with respect to the number of jobs.

- we demonstrates an average improvement of 11% over existing state-of-the-art methods under comparable runtime budgets. Moreover, S&S maintains high-quality schedules as problem size increases, highlighting its robustness for large-scale applications.

## 2 RELATED WORKS

Mascis & Pacciarelli (2002) conducted one of the earliest comprehensive studies on BJSP, introducing the widely adopted Alternative Graph model as well as a heuristic algorithm. Later work for BJSP can be broadly divided into exact and approximate methods. Exact methods are primarily based on Branch-and-Bound (B&B). AitZai et al. (2012); Dabah et al. (2018; 2016) attempted to obtain exact solutions via B&B, further exploring parallel acceleration using physical hardware. More recently, Rihane et al. (2022) innovatively incorporated learning-based techniques to reduce search time, where efficient learning of branching and selection strategies significantly sped up the process, achieving near state-of-the-art performance with substantially fewer search iterations. Although exact methods guarantee optimality, their scalability is severely limited due to the intrinsic complexity of BJSP, which makes them impractical for real-world scenarios requiring high-quality solutions within limited computational resources.

Approximate methods for the BJSP are mainly based on metaheuristic approaches, which aim to obtain high-quality solutions without guaranteeing exact optimality. Among these, Tabu Search (TS) (Glover, 1990) has been the most extensively studied. TS is a local search–based optimization method that avoids cycling by prohibiting recently visited solutions, thereby escaping local optima and exploring broader solution spaces. However, unlike JSP, where classical neighborhoods (N1, NA, NB, N2, N4, N5; (Błażewicz et al., 1996)) are effective, applying them to BJSP tends to yield a high proportion of infeasible solutions (Mogali et al., 2021) . Repairing such solutions incurs significant computational cost. To mitigate this, Gröflin & Klinkert (2009) attempted to directly construct feasible neighborhoods, while Dabah et al. (2017) proposed heuristic reconstruction strategies to improve solution quality. Dabah et al. (2019) further introduced a parallel multistart approach to accelerate the time-consuming repair process, leveraging 512-core hardware to speed up the search. More recently, Luo et al. (2021) argued that not all neighbors contribute meaningfully to the search and developed theoretical insights to reduce complexity, leading to significant efficiency improvements and achieving the first solution for instances of size 100×20. Beyond TS, other metaheuristics have also been investigated, such as Iterated Greedy, Simulated Annealing, and their variants (Lange & Werner, 2019a; Pranzo & Pacciarelli, 2016; van Blokland, 2012; Lange & Werner, 2019b) . Metaheuristics generally produce high-quality solutions on small and medium size instances and benefit from well-designed heuristics. However, they rely on manually crafted neighborhoods and often require many iterations, which results in long computational times and limits their scalability and responsiveness in dynamic or large-scale scheduling scenarios.

## 3 PRELIMINARIES

**BJSP** Let $\mathcal{J} = \{1, \ldots, n\}$ be the set of jobs and $\mathcal{M} = \{1, \ldots, m\}$ the set of machines. Each job $j \in \mathcal{J}$ is an ordered sequence of operations $\mathcal{O}_j = (o_{j,1}, \ldots, o_{j,n_j})$. Operation $o_{j,k}$ requires machine $m_{j,k} \in \mathcal{M}$ and has processing time $p_{j,k} > 0$.A schedule (solution) is specified by starting times $S = \{s_{j,k}\}_{1 \leq j \leq n, 1 \leq k \leq n_j}$ with $s_{j,k} \in \mathbb{N}$. The completion time is $c_{j,k} := s_{j,k} + p_{j,k}$.In the classical Job Shop Scheduling Problem (JSP), once an operation finishes its processing, the machine is immediately released. In contrast, in the Blocking Job Shop Scheduling Problem (BJSP), no intermediate buffers are available between machines. Therefore, if the successor machine of an operation is occupied, the job remains on its current machine after completion, thereby blocking the machine until the next machine becomes free (Mascis & Pacciarelli, 2002).The size of a BJSP instance is denoted as $|\mathcal{J}| \times |\mathcal{M}|$. Mathematical formulation about BJSP can be found in A.2

**Submodular** Let $N$ be a ground set. Any function $f : 2^N \to \mathbb{R}$ is called a *set function*. A set function $f$ is *submodular* if, for any $A \subseteq B \subseteq N$ and any $v \notin B$, it holds that

$$f(A \cup \{v\}) - f(A) \geq f(B \cup \{v\}) - f(B). \tag{1}$$

A common problem concerning submodular functions is *Cardinality-Constrained submodular Maximization* (Nemhauser et al., 1978). Formally, given a submodular function $f : 2^{\mathcal{N}} \to \mathbb{R}$ and a cardinality constraint $k$, the goal is to find a subset $S \subseteq \mathcal{N}$ with $|S| \leq k$ that maximizes $f(S)$:

$$\max_{S \subseteq \mathcal{N},\, |S| \leq k} f(S). \tag{2}$$

Submodular functions exhibit structural properties that allow greedy algorithms to be equipped with provable and often tight approximation guarantees. A submodular function $f$ is said to be *monotone* if its value never decreases when elements are added to the set,it holds that:

$$f(A) \leq f(B), \quad \forall A \subseteq B \subseteq \mathcal{N}. \tag{3}$$

For the problem of maximizing a monotone submodular function subject to a cardinality constraint, the classical greedy algorithm that iteratively selects the element with the largest marginal gain achieves a $(1-1/e)$-approximation ratio (Nemhauser et al., 1978). In contrast, for the non-monotone case, the best-known algorithms currently guarantee a $0.377$-approximation ratio (Chen et al., 2024).

## 4 METHOD

The iterative procedure of our method is illustrated in Figure 2 and can be divided into three main stages: Selection, Scheduling, and Schedule Retention. In the Selection stage, a subset of jobs is chosen from the candidate pool and added to the set of jobs currently in process. During the Scheduling stage, these in-process jobs are scheduled in detail using the lower-level solver. In the Schedule Retention stage, a portion of the resulting schedule is preserved, while completed jobs are removed from the in-process set. This cycle repeats iteratively until all jobs are fully scheduled, constructing a complete schedule while maintaining a manageable problem size at each step.

**Selection Process.** Our method relies on a valuation network to assess whether a group of jobs exhibits good parallelizability. For instance, in the BNS scenario illustrated in Figure 1, `job1` and `job2` can partially execute in parallel, whereas `job3` must run serially with `job2`. To quantify parallelism within a job group, we adopt machine utilization—defined as the total machine processing time divided by the makespan—as the evaluation metric. This choice is motivated by two factors: first, maximizing machine utilization directly aligns with our optimization objective, as the total processing time is fixed and higher utilization implies a shorter makespan. More importantly, we make the novel observation that the machine utilization of a job set often exhibits approximate submodular behavior: adding a job to a relatively small set typically increases utilization more than adding it to an almost full set. This near-submodularity provides a principled abstraction for guiding our selection process and motivates the application of submodular optimization techniques. Empirical evidence in Appendix A.4 demonstrates that this property holds in a large proportion of practical instances, highlighting its practical relevance and validating our approach.

During selection, we greedily pick jobs that maximize the total machine utilization at each step. Selecting $k$ elements from the full set $\mathcal{N}$ to form a subset $S$ that maximizes the set function $f_\theta(S)$ is a challenging problem. However, the submodular property of $f_\theta$ provides a theoretical foundation and performance guarantee. In most cases, when $S$ already contains a moderate number of jobs, it suffices to greedily select a single job $a$ that maximizes $f_\theta$:

$$a = \arg \max_a f_\theta(S \cup \{a\}). \tag{4}$$

In other cases, we employ the *Guided Maximal Combinatorial Choice (GMCC)* algorithm (Chen et al., 2024). GMCC introduces a guided randomized greedy framework that surpasses the $1/e$ approximation barrier for constrained non-monotone submodular maximization. It first applies a fast local search to construct a *guidance set $Z$* that captures suboptimal regions, and then runs a modified randomized greedy algorithm leveraging $Z$ to steer the selection. Detailed algorithmic steps are provided in the Appendix A.3 and in Chen et al. (2024).

**Scheduling** In our framework, we employ a **Constraint Programming (CP) solver** as the sub-solver to optimize the scheduling of the selected job subset. CP is a widely used paradigm for combinatorial optimization (Li et al., 2025), which models problems in terms of variables, domains, and constraints, and efficiently searches for feasible solutions that satisfy all constraints. Using a CP

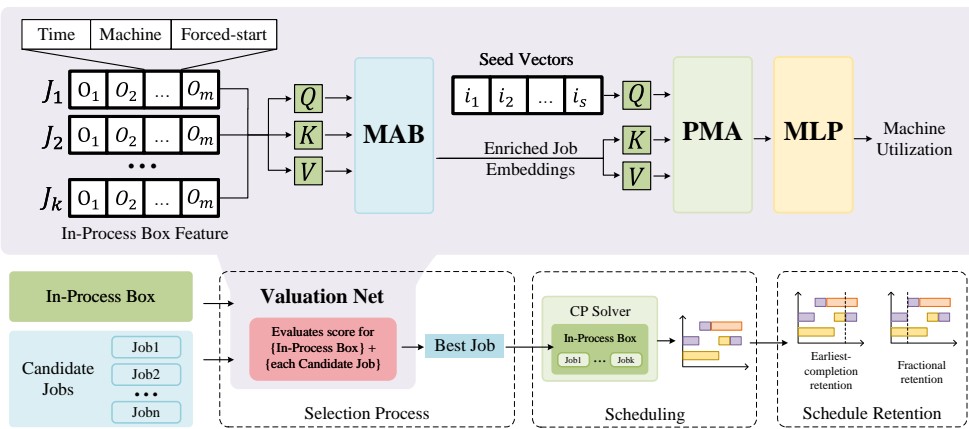

Figure 2: Our hierarchical optimization framework and the detailed architecture of the Valuation Net. The lower panel shows one iteration of our method, which has three main stages. First, in the Selection Process, the Valuation Net evaluates each job in the Candidate Jobs pool, and the job with the highest score is chosen as the Best Job. Second, in the Scheduling stage, this best job is added to the In-Process Box, and the CP Solver then creates a schedule for this complete job subset. Finally, in the Schedule Retention stage, we use a policy to decide which part of the schedule to keep. The upper panel shows the specific structure of the Valuation Net , a Set Transformer model. Input job features are first processed by a Multihead Attention Block (MAB) to capture dependencies between jobs and generate enriched job embeddinsgs. Next, a Pooling by Multihead Attention (PMA) module uses learnable Seed Vectors to aggregate these job embeddings into a fixed-size vector. Finally, this vector is passed to an MLP to predict the overall Machine Utilization.

solver as our subsolver offers several benefits. First, it allows our framework to leverage a mature and general-purpose solver, reducing implementation complexity. Second, CP is highly extensible, which means our framework can be readily adapted to solve variations of the problem, such as the Blocking Job Shop Scheduling Problem with finite waiting times or other practical extensions.

**Schedule Retention.** At each iteration, we apply a *schedule retention* strategy to decide which portion of the solver-generated schedule $\sigma$ should be kept and appended to the global solution $\Pi$. Concretely, we choose a time $\tau$ and retain all operations scheduled before $\tau$, including those already started but not yet completed at $\tau$, while discarding all decisions beyond $\tau$. For truncated jobs, the partially executed operations are treated as already committed: in the next iteration, such operations are forced to start at time 0 on their assigned machines, with their processing times reduced to reflect the portion already executed. Completed operations are removed, and fully finished jobs are discarded. The preserved part $\sigma_{\text{retain}}$ is appended to the global schedule $\Pi$, and the remaining subproblem is updated for the next iteration. We proposed two retention strategy : **Earliest-completion retention.** The time point $\tau$ is chosen as the earliest completion among all operations in $\sigma$. This selection is motivated by the observation that, immediately after $\tau$, the set of in-process jobs is guaranteed to change—often accompanied by the addition of new jobs—making scheduling beyond this point less informative. Moreover, by limiting the schedule to $\tau$, we avoid redundant computations arising from overlapping subproblems at earlier time points, thus improving efficiency. **Fractional retention.** For large-scale subproblems or limited solver budgets, $\tau$ is set to a fixed fraction of the earliest completion time. The overlap between consecutive subproblems in our approach is analogous to the rolling-horizon optimization method in Li et al. (2025), allowing repeated subproblem solutions to improve overall quality even when individual solver calls are not optimal.

**Network Architecture.** To model a set of $n$ jobs, each with $m$ operations, we adopt the **Set Transformer** architecture (Lee et al., 2019). There are two main motivations for this choice. First, we need a network that naturally operates on sets, as the order of jobs should not affect the prediction.

Second, the operational features of a single job— the machine IDs of its operactions —are often not meaningful in isolation. Their impact becomes apparent only in relation to other jobs, when multiple operations require the same machine. Set-based attention allows the model to capture these inter-job dependencies effectively.

Let $x_{jk} = [p_{jk}, m_{jk}, t_{jk}]$ denote the feature vector of the $k$-th operation of the $j$-th job where $p_{jk}$ is the processing time, $m_{jk}$ is the machine ID, and $t_{jk}$ indicates whether the operation is forced to start. Each operation is embedded as

$$\mathbf{o}_{jk} = [\text{Linear}(p_{jk}); \text{Embed}(m_{jk}); \text{Embed}(t_{jk})] \in \mathbb{R}^{d_o}, \tag{5}$$

where $d_o = d_p + d_m + d_t$ is the dimension of the concatenated embedding. A learnable positional encoding $\mathbf{s}_k \in \mathbb{R}^{d_o}$ is added to $\mathbf{o}_{ijk}$ to encode the operation's position within the job. The embedding of the $j$-th is obtained by flattening all $m$ operation embeddings:

$$\mathbf{j}_j = \text{Flatten}(\mathbf{o}_{j1}, \ldots, \mathbf{o}_{jm}) \in \mathbb{R}^{m \cdot d_o}. \tag{6}$$

The $n$ job embeddings form a set $\mathbf{J} = \{\mathbf{j}_1, \ldots, \mathbf{j}_n\}$. We pass $\mathbf{J}_i$ through $L$ layers of Set Attention Blocks(SAB). Each SAB performs multi-head self-attention followed by a feed-forward network (FFN) with residual connections and layer normalization:

$$\mathbf{H} = \text{LayerNorm}(\mathbf{J} + \text{MultiheadAttention}(\mathbf{J}, \mathbf{J}, \mathbf{J})) \tag{7}$$

$$\mathbf{J}' = \text{LayerNorm}(\mathbf{H} + \text{FFN}(\mathbf{H})). \tag{8}$$

Here, $\text{MultiheadAttention}(Q, K, V)$ denotes the multi-head attention (Vaswani et al., 2017)with query $Q$, key $K$, and value $V$, and FFN is a two-layer MLP with ReLU activation. This mechanism allows each job to attend to other jobs in the set $\mathcal{J}$. After $L$ SAB layers, we aggregate the set into a fixed-size vector using Pooling by Multihead Attention (PMA):

$$\mathbf{z} = \text{PMA}(\mathbf{J}^{(L)}) = \text{Concat}(\mathbf{A}_1\mathbf{J}^{(L)}, \ldots, \mathbf{A}_s\mathbf{J}^{(L)})W \in \mathbb{R}^{s \cdot md_o}, \tag{9}$$

where $s$ is the number of learnable seed vectors $\mathbf{S} = [\mathbf{s}_1, \ldots, \mathbf{s}_s] \in \mathbb{R}^{s \times d_o}$, and each attention map is computed as

$$\mathbf{A}_i = \text{softmax}\left(\frac{\mathbf{s}_i(\mathbf{J}^{(L)})^\top}{\sqrt{d_o}}\right) \in \mathbb{R}^{1 \times n}. \tag{10}$$

Each seed vector attends to all jobs in the set to summarize set-level information, producing $\mathbf{z}_i = \mathbf{A}_i\mathbf{J}^{(L)}$, and the outputs of all seeds are concatenated and optionally projected by $W$.The pooled vector $\mathbf{z}$ is then mapped to a scalar prediction via an MLP:

$$\hat{y} = \text{MLP}(\mathbf{z}) \in \mathbb{R}, \tag{11}$$

where $\hat{y}$ denotes the predicted average machine utilization. This design naturally handles variable-sized job sets, preserves permutation invariance, and captures inter-job dependencies through attention.

**Training**  We adopt a supervised learning approach using randomly generated BJSP instances. To simulate realistic scheduling scenarios, we apply two types of perturbations to the job sets: (i) randomly masking a subset of operations, which mimics partially executed jobs, and (ii) randomly removing completed jobs. The CP solver is then used to compute supervision labels, defined as the *machine utilization ratio*, which reflects the degree of parallelism among jobs. For large instances where the CP solver cannot reach optimality within a reasonable time, we use the best solution found within a fixed time cutoff as the target. The network is trained to minimize the mean squared error (MSE) between the predicted and target utilization values.

## 5 EXPERIMENTS

Our framework is mainly designed for large-scale BJSP instances. In standard scenarios, inference follows the three-step procedure described above. However, when the number of jobs is close to the number of machines, almost all jobs can be in process simultaneously, making the selection step

redundant. In such low-dimensional cases, our method naturally reduces to directly applying the second schedule-preserving strategy without invoking the network-based selection.

We evaluate our proposed framework(S&S) against several baselines on standard benchmarks and large-scale synthetic instances. More detailed experiments can be found in the appendix A.5. Our study is guided by these research questions: How well does S&S perform on large-scale BJSP instances, including extreme sizes? How our proposed framework performs in general scenarios? How much benefit does the learned selection network provide compared to random or oracle-based strategies?

**Datasets** We conduct experiments on both public benchmarks and synthetically generated instances. Specifically, we evaluate our framework on the Lawrence instances (Lawrence, 1984) and Taillard instances (Taillard, 1993), which are standard testbeds for job shop scheduling. To further assess scalability beyond existing benchmarks, we additionally construct larger synthetic instances using the widely adopted Taillard generation procedure (Taillard, 1993), with sizes reaching up to $(1000, 20)$. In total, our study spans problem sizes of up to 20,000 operations, substantially extending the scale considered in prior BJSP research. For comparison, most previous works were limited to fewer than 600 operations, while Mogali et al. (2021) was the first to give results on instances approaching 2000 operations.

**Baselines** We compare against the following methods: **Tabu Search:** the current state-of-the-art algorithm for BJSP, employing the $N4/N5$ neighborhood structures (Mogali et al., 2021). This solver is widely regarded as the strongest heuristic for BJSP to date, and has established the best known solutions for nearly all benchmark instances considered in our study. **CP Solver:** a widely used exact solver. **R-S&S:** a variant of our method without the network, where jobs are selected randomly but with the same selection procedure.

**Implementation Details** We describe the experimental setup and hyperparameters used throughout training. The model employs 16-dimensional embeddings for machines and processing times, and a 4-dimensional embedding for the forced-start flag. The hidden dimension is set to 64, with four attention heads and six stacked attention layers. Training instances are generated by perturbing 1,000 randomly created BJSP instances: each job is removed with probability 0.03, and, for surviving jobs, an operation along with all its predecessors is removed with probability 0.2. The model is trained for 1,000 epochs with a learning rate of 0.001, using 10% of the data for testing.

For synthetic evaluation, 100 instances are generated for each problem configuration, except for very large instances $(500, 20)$ and $(1000, 20)$, which are limited to 10 instances due to computational cost. In our hierarchical framework, large-scale problems ($n \geq 50$) use an Earliest-completion retention, while smaller problems or those where the number of jobs is close to the number of machines employ a Fractional retention. For the CP solver baseline, subproblems that are too large to solve exactly are limited to 50 seconds in the unlimited setting. When a global time constraint is imposed, the allocated time for each subproblem is approximately the total runtime budget divided by the number of jobs.

Table 1: Comparison of Tabu Search and S&S on TA instances

| Instance | Size | Tabu 60 Avg Obj | S&S 60 Obj | Gap (%) | Tabu 1800 Avg Obj | S&S 1800 Obj | Gap (%) |
|---|---|---|---|---|---|---|---|
| TA71 | 100×20 | 17426.6 | 14895 | -14.53% | 12369.4 | 12285 | -0.68% |
| TA72 | 100×20 | 16225.8 | 15763 | -2.85% | 11745.6 | 12534 | 6.71% |
| TA73 | 100×20 | 17370.4 | 15313 | -11.84% | 12078.6 | 12358 | 2.31% |
| TA74 | 100×20 | 16963.9 | 14788 | -12.83% | 12044.8 | 13067 | 8.49% |
| TA75 | 100×20 | 17127.6 | 15151 | -11.54% | 11911.4 | 12156 | 2.05% |
| TA76 | 100×20 | 16578.0 | 14774 | -10.88% | 12223.8 | 12321 | 0.80% |
| TA77 | 100×20 | 17674.8 | 16365 | -7.41% | 12412.2 | 12511 | 0.80% |
| TA78 | 100×20 | 17007.8 | 15014 | -11.72% | 11898.6 | 12807 | 7.63% |
| TA79 | 100×20 | 17145.8 | 15834 | -7.65% | 12118.4 | 12250 | 1.09% |
| TA80 | 100×20 | 16186.4 | 15100 | -6.71% | 11729.0 | 11825 | 0.82% |

**Results on Benchmark.** Table 1 presents a comparison between our model and the state-of-the-art Tabu Search on large-scale benchmarks. In our experiments, we considered two scenarios that correspond to practical settings. The first scenario represents a dynamic environment where a reasonable solution must be obtained within a very short time. We set the runtime limit to 60 seconds. The second scenario represents a static environment where sufficient but reasonable time is available to obtain the best possible solution, for which we set the runtime limit to 1800 seconds.

Under the 60-second setting, our method achieves consistently better results than the baseline across all datasets, with an average improvement of 11%, demonstrating the high computational efficiency of our approach on large-scale problems. Under the 1800-second setting, while our method falls slightly behind in some cases, in most instances the gap is within 2%, essentially reaching the same best performance as the state of the art. These results validate both the efficiency and the solution quality of our method.

Table 2 reports the performance of S&S on the small-scale LA benchmark, representing a secondary scenario where the small problem size reduces the impact of the pre-selection strategy. Overall, S&S achieves strong early-stage performance: under the 60-second budget, it matches or slightly outperforms Tabu Search on most instances. With a longer 600-second budget, it generally attains solution quality comparable to Tabu Search, though in some less favorable instances, a performance gap remains. These results demonstrate that even in disadvantageous scenarios, S&S maintains highly efficient early-stage optimization while remaining broadly competitive with state-of-the-art solvers. Complete results for all instances in Ta and La are provided in Appendix A.5.

Table 2: Comparison between Tabu Search and S&S on LA instances

| Instance | Size | Tabu 60 Avg Obj | S&S 60 Obj | Gap (%) | Tabu 1800 Avg Obj | S&S 1800 Obj | Gap (%) |
|---|---|---|---|---|---|---|---|
| LA01–LA05 | 10*5 | 836 | 836.4 | 0.05% | 836 | 836.4 | 0.05% |
| LA06–LA10 | 15*5 | 1212.94 | 129.6 | 1.36% | 1203.4 | 1220.8 | 1.43% |
| LA11–LA15 | 20*5 | 1554.44 | 1569.4 | 0.89% | 1494.2 | 153.2 | 3.42% |
| LA16–LA20 | 10*10 | 1085.5 | 1082.6 | -0.26% | 1082.6 | 1082.6 | 0.00% |
| LA21–LA25 | 15*10 | 1481.5 | 1450.8 | 1.84% | 1418.8 | 1494.4 | 5.33% |
| LA26–LA30 | 20*10 | 1997.0 | 2075.2 | 5.35% | 1888.6 | 2060 | 9.08% |
| LA31–LA35 | 30*10 | 2927.32 | 3298 | 12.66% | 2777 | 3091.8 | 11.34% |
| LA36–LA40 | 15*15 | 1809.2 | 1804.8 | -0.25% | 1727.2 | 1767.6 | 2.34% |

Table 3: Comparison of S&S, Random S&S, and CP solver across different problem sizes

| Mac Num | Job Num | S&S Avg Obj | S&S Time (s) | Random S&S Avg Obj | Random S&S Time (s) | CP solver Avg Obj | CP solver Time (s) |
|---|---|---|---|---|---|---|---|
| 5 | 100 | 9056.1 | 25.2 | 9124.94 | 6.7 | 8660 | 3600 |
| | 200 | 18005.7 | 92.3 | 18162.1 | 12.7 | 27148 | 7200 |
| | 500 | 44752.7 | 555.2 | 45310.7 | 34.3 | - | - |
| | 1000 | 89398.1 | 1996.1 | 90380.45 | 74.7 | - | - |
| 10 | 100 | 10512.2 | 428.7 | 10522.16 | 398.7 | 21473 | 3600 |
| | 200 | 20850.3 | 1031.2 | 20981.1 | 883.4 | 73253 | 7200 |
| | 500 | 52046.9 | 2846.0 | 52088.1 | 2307.7 | - | - |
| | 1000 | 103811.2 | 5600.0 | 104032.4 | 3676.8 | - | - |
| 20 | 100 | 14358.3 | 4340.3 | 14446.8 | 4300.5 | 72135 | 3600 |
| | 200 | 27152.2 | 9447.4 | 27284.4 | 9365.3 | - | - |
| | 500 | 66565.8 | 25103.4 | 66671.8 | 24600.3 | - | - |
| | 1000 | 133597.7 | 52293.5 | 134730.5 | 50186.5 | - | - |

**Results on Larger-Scale instances.** Table 3 reports the performance of S&S on large-scale BJSP instances generated from our production dataset, involving up to 20 machines and 1000 jobs. For each instance, we compare S&S with the Random S&S baseline, where job subsets are selected randomly without the valuation network. Across all tested scenarios, S&S consistently achieves lower objective values than Random S&S. This indicates that the network-based selection effectively identifies job subsets with higher parallelizability, allowing the solver to produce schedules with

better machine utilization. The performance gap between S&S and Random S&S increases with problem size, suggesting that the network contributes more substantially as the instance grows.

On moderately large instances, the standalone CP solver yields worse objective values even with more computation time, and for the largest instances it fails to produce solutions due to memory limitations, illustrating the inherent difficulty of BJSP. Notably, our method also employs a CP solver as the internal optimizer. The performance difference arises because, within the S&S framework, the CP solver is applied to a carefully selected subset of jobs rather than the full problem. This restricted formulation substantially reduces the search space, enabling the CP solver to operate effectively where it would otherwise fail. These results suggest that the strength of S&S lies in the interaction between learning-based selection and CP optimization, rather than in CP alone.

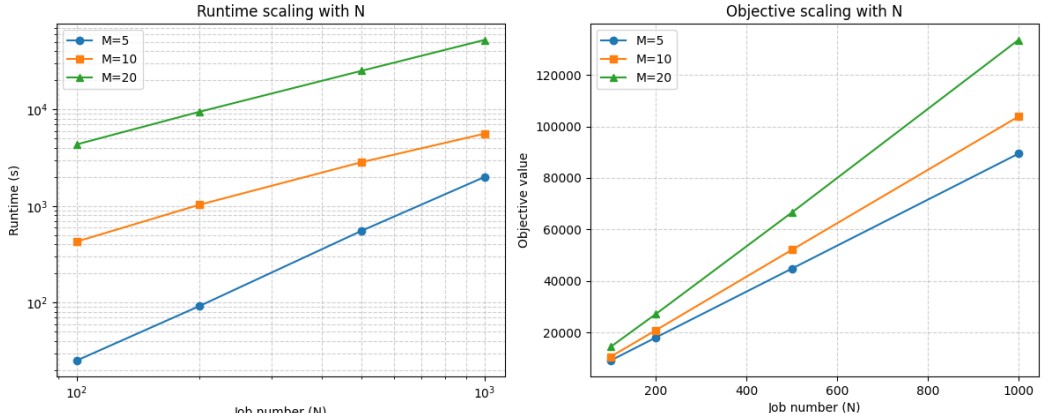

Figure 3: Scaling behavior of our method on synthetic BJSP instances. (**Left**) Runtime grows nearly linearly with the number of jobs $N$ under different machine settings ($M = 5, 10, 20$). (**Right**) The obtained objective values also scale linearly with $N$.

To further evaluate the scalability of S&S, we examine the relationship between problem size, runtime, and solution quality. Figure 3 shows that, for a fixed number of machines, the runtime of S&S increases approximately linearly with the number of jobs. Correspondingly, the solution objective also grows roughly linearly with the job count. This linear trend aligns with the intuition that, under the fixed data-generation distribution, the expected total processing time increases proportionally with the number of jobs. The observed linear scaling indicates that the computational complexity of S&S grows moderately with problem size, and that the method remains effective when extrapolated to very large instances. In contrast, CP solvers fail to produce solutions for the largest instances due to memory constraints, and their solution quality deteriorates even on moderately sized problems. These results suggest that S&S maintains both computational efficiency and high-quality scheduling performance across a wide range of problem sizes, highlighting its practical applicability for large-scale BJSP scenarios.

## 6 CONCLUSION

This paper introduces Select and Schedule (S&S), a hierarchical optimization framework for the Blocking Job Shop Scheduling Problem (BJSP) that scales efficiently to large instances. By exploiting the observation that blocking constraints limit concurrent jobs to the number of machines, S&S uses a high-level neural network to select a subset of jobs, which are then scheduled by a lower-level CP solver. Extensive experiments on standard and large-scale benchmarks show that S&S consistently produces high-quality solutions. Under tight time constraints (e.g., 60 seconds), it outperforms state-of-the-art Tabu Search, while remaining competitive with longer time budgets (1800 seconds). S&S demonstrates robustness and efficiency even for extremely large instances, offering a practical and scalable solution for real-world dynamic scheduling.

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

# A  APPENDIX

## A.1  USE OF LARGE LANGUAGE MODELS

During the preparation of this work, we employed a Large Language Model (LLM) to polish English descriptions, improving clarity, grammar, and academic style, as well as to provide guidance in generating text-based prompts for schematic figures, illustrations, and algorithmic diagrams. All substantive technical decisions, experimental design, core algorithmic code were made by the authors; the use of the LLM served solely as an auxiliary tool to enhance presentation, and visualization. We carefully verified the outputs produced with LLM assistance and are fully responsible for the correctness and integrity of all results and claims presented in this work.

## A.2  BJSP MATHEMATICAL FORMULATION

The mathematical formulation of BWS is as follows. These constraints are directly incorporated into our CP solver, ensuring that both the technological order and machine capacity with blocking are strictly enforced during the scheduling process.

$$s_{j,k+1} \geq s_{j,k} + p_{j,k}, \quad \forall 1 \leq j \leq n, \ 1 \leq k \leq n_j. \tag{12}$$

$$s_{j,k} \geq s_{j',k'+1} \quad \text{or} \quad s_{j',k'} \geq s_{j,k+1},$$
$$\forall j, j', \ 1 \leq j, j' \leq n, \ 1 \leq k < n_j, \ 1 \leq k' < n_{j'}, \ m_{j,k} = m_{j',k'}, j \neq j'. \tag{13}$$

$$s_{j,k} \geq s_{j',k'+1} \quad \text{or} \quad s_{j',k'} \geq s_{j,k} + p_{j,k},$$
$$\forall j, j', \ 1 \leq j, j' \leq n, \ k = n_j, \ 1 \leq k' < n_{j'}, \ m_{j,k} = m_{j',k'}, j \neq j'. \tag{14}$$

In the above formulation, $s_{j,k}$ denotes the start time of the $k$-th operation of job $j$, and $p_{j,k}$ represents its processing time. The first equation enforces the technological order: each operation must start only after its preceding operation is completed. The second constraint corresponds to the general blocking condition: for any two operations sharing the same machine, at least one must start only after the successor of the other has begun. The third constraint captures the special case where an operation is the last one of its job; since it has no successor, its completion immediately releases the machine. The above formulation corresponds to the Blocking No-Wait Shop (BNS) problem. For the Blocking Job Shop (BWS) problem, Equations (2) and (3) are slightly modified as follows.

$$s_{j,k} > s_{j',k'+1} \quad \text{or} \quad s_{j',k'} > s_{j,k+1},$$
$$\forall j, j', \ 1 \leq j, j' \leq n, \ 1 \leq k < n_j, \ 1 \leq k' < n_{j'}, \ m_{j,k} = m_{j',k'}, j \neq j'. \tag{15}$$

$$s_{j,k} > s_{j',k'+1} \quad \text{or} \quad s_{j',k'} \geq s_{j,k} + p_{j,k},$$
$$\forall j, j', \ 1 \leq j, j' \leq n, \ k = n_j, \ 1 \leq k' < n_{j'}, \ m_{j,k} = m_{j',k'}, j \neq j'. \tag{16}$$

## A.3  GUIDED MULTI-STAGE GREEDY COMBINATORIAL ALGORITHM

The GMGC algorithm (Guided Multi-stage Greedy Combinatorial) is illustrated in Algorithm 1. The main symbols used are as follows: $\mathcal{U}$ denotes the ground set of elements, $f : 2^{\mathcal{U}} \to \mathbb{R}$ is the submodular objective function, $\mathcal{I}$ represents the constraints (e.g., cardinality or matroid constraints), $k$ is the selection budget, $Z_0$ is the initial approximate solution, $Z$ is the guidance set generated in the first stage, $A$ is the solution obtained in the second-stage randomized greedy selection, $t \in [0, 1]$ is the switching ratio controlling the number of initial steps that exclude elements in the guidance set, and $\epsilon$ is the precision parameter used to set the marginal gain threshold during guidance set construction.

In the first stage, the guidance set $Z$ is constructed via the FASTLS subroutine. Elements are iteratively added or replaced in $Z$ only if the improvement in marginal gain exceeds the threshold $\epsilon/k \cdot f(Z)$ and the resulting set satisfies the constraints $\mathcal{I}$. This process continues until no further improvement is possible, yielding a guidance set that provides structural information and quality guarantees for subsequent selection.

In the second stage, the GUIDEDRG subroutine performs a randomized greedy selection. During the first $t \cdot k$ steps, elements from the guidance set are excluded to exploit its structure, while in the

---

**Algorithm 1:** GMGC (Guided Multi-stage Greedy Combinatorial) Algorithm

---

**Input:** Submodular function $f : 2^{\mathcal{U}} \to \mathbb{R}$, constraint $\mathcal{I}$, initial solution $Z_0$, accuracy $\epsilon$, budget $k$, switching ratio $t$.

**Output:** Final solution $S$.

1 **Phase 1: Guided Set Construction (FASTLS)**
2 Initialize $Z \leftarrow Z_0$;
3 **repeat**
4     **foreach** $a \in \mathcal{U}$ **do**
5         **if** $a \in Z$ **then**
6             **foreach** $e \in \mathcal{U} \setminus Z$ **do**
7                 **if** $Z' = (Z \setminus \{a\}) \cup \{e\} \in \mathcal{I}$ **and**
                    $\Delta(e \mid Z \setminus \{a\}) - \Delta(a \mid Z \setminus \{a\}) \geq \frac{\epsilon}{k} \cdot f(Z)$ **then**
8                     $Z \leftarrow Z'$; break;
9         **else**
10             **if** $\Delta(a \mid Z) = f(Z \cup \{a\}) - f(Z) \geq \frac{\epsilon}{k} \cdot f(Z)$ **then**
11                 $Z \leftarrow Z \cup \{a\}$;

12 **until** *no improvement*;
13 **Phase 2: Guided Randomized Greedy (GUIDEDRG)**
14 Initialize $A \leftarrow \emptyset$;
15 **for** $i \leftarrow 1$ **to** $t \cdot k$ **do**
16     Compute $\Delta(u \mid A)$ for all $u \in \mathcal{U} \setminus Z$;
17     Let $M_i$ be the set of top-$r$ elements by marginal gain, where
        $r = \min(k - |A|, t \cdot k - |A|)$;
18     Pick $x_i$ uniformly at random from $M_i$;
19     $A \leftarrow A \cup \{x_i\}$;
20 **for** $i \leftarrow t \cdot k + 1$ **to** $k$ **do**
21     Compute $\Delta(u \mid A)$ for all $u \in \mathcal{U}$;
22     Let $M_i$ be the set of top-$r$ elements by marginal gain, where $r = k - |A|$;
23     Pick $x_i$ uniformly at random from $M_i$;
24     $A \leftarrow A \cup \{x_i\}$;

25 **Final Selection:**
26 Return $S = \arg\max\{f(Z), f(A)\}$;

---

remaining $k - t \cdot k$ steps, all elements in the ground set are considered. At each step, a candidate pool is formed by selecting elements with the largest marginal gains, and one element is chosen uniformly at random to be added to the current solution $A$. Finally, the algorithm compares the objective values of the guidance set $Z$ and the greedy solution $A$, and returns the one with the higher value as the final output.

This two-stage design leverages the high-quality structure of the guidance set while retaining the exploratory power of randomized greedy selection, achieving strong theoretical guarantees and practical performance.

## A.4 SUBMODULAR DISCUSSION

As discussed in the main text, the utilization function $F(S)$ is not strictly submodular in all cases. For instance, consider sets

$$A = \{[(1, m_1), (1, m_2)]\}, \quad B = \{[(1, m_1), (1, m_2)], [(1, m_1), (98, m_2)]\},$$

and a new job

$$c = [(98, m_1), (1, m_2)].$$

We have $F(A) = 1$, $F(B) = 1.01$, $F(A \cup \{c\}) = 1.01$, and $F(B \cup \{c\}) \approx 1.98$, so the marginal gain is larger for the superset $B$, violating strict submodularity.

Nevertheless, we conducted an empirical study to verify that $F(S)$ exhibits approximate submodular behavior in most cases(t) when the total number of jobs in a set does not exceed the number of machines $m$. The experimental procedure is as follows:

1. Generate random instances of the blocking job shop problem with up to $n$ jobs and $m$ machines.

2. For each instance, randomly construct a subset $A$ of jobs with size up to $m$, and a superset $B \supseteq A$ with size up to $m$.

3. Sample a new random job $c$.

4. Solve the scheduling problem for $A$, $B$, $A \cup \{c\}$, and $B \cup \{c\}$ to obtain their machine utilizations $F(A)$, $F(B)$, $F(A \cup \{c\})$, $F(B \cup \{c\})$.

5. Check whether the marginal gain satisfies

$$F(A \cup \{c\}) - F(A) \geq F(B \cup \{c\}) - F(B).$$

6. Repeat steps 2–5 for a large number of trials (e.g., 10,000) and record the proportion of cases satisfying the inequality.

Based on over 10,000 tests with $m = 5$ and $m = 7$, we find that the utilization function satisfies the submodularity inequality in the vast majority of cases(almost 100%). This provides strong empirical support for treating it as approximately submodular in our framework.

## A.5 DETAIL RESULTS ON BENCHMARK

This is all the test results we have on the Ta and La datasets. Table 4 6 reports the full experimental results on the LA and TA benchmark sets, covering instances of varying sizes from small to extremely large. Overall, our method consistently achieves solutions close to or surpassing those of Tabu Search across all instances. Under short time budgets , S&S demonstrates a clear advantage, especially on large-scale TA instances, where it rapidly converges to high-quality solutions, showcasing strong early-stage optimization ability. With longer budgets, S&S remains highly competitive: while Tabu Search occasionally achieves slightly better results on medium-scale cases, the gap is negligible, and S&S frequently matches or outperforms it. These results confirm that S&S effectively combines fast convergence with robust scalability, making it not only competitive with state-of-the-art metaheuristics but also a practical solution for real-world dynamic scheduling applications.

Table 4: Comparison between Tabu Search and S&S on LA instances

| Instance | Size | 60s | | | 600s | | |
|---|---|---|---|---|---|---|---|
| | | Tabu Obj | S&S Obj | Gap (%) | Tabu Obj | S&S Obj | Gap (%) |
| LA01 | 10*5 | 881 | 881 | 0.00% | 881 | 881 | 0.00% |
| LA02 | 10*5 | 900 | 900 | 0.00% | 900 | 900 | 0.00% |
| LA03 | 10*5 | 808 | 810 | 0.25% | 808 | 810 | 0.25% |
| LA04 | 10*5 | 859 | 859 | 0.00% | 859 | 859 | 0.00% |
| LA05 | 10*5 | 732 | 732 | 0.00% | 732 | 732 | 0.00% |
| LA06 | 15*5 | 1203.2 | 1214 | 0.90% | 1194 | 1194 | 0.00% |
| LA07 | 15*5 | 1132.1 | 1129 | -0.27% | 1127 | 1127 | 0.00% |
| LA08 | 15*5 | 1190.2 | 1189 | -0.10% | 1173 | 1173 | 0.00% |
| LA09 | 15*5 | 1312.3 | 1311 | -0.10% | 1305 | 1305 | 0.00% |
| LA10 | 15*5 | 1226.9 | 1305 | 6.37% | 1218 | 1305 | 7.14% |
| LA11 | 20*5 | 1588.2 | 1605 | 1.06% | 1501 | 1590 | 5.93% |
| LA12 | 20*5 | 1414.1 | 1396 | -1.28% | 1353 | 1396 | 3.18% |
| LA13 | 20*5 | 1545.6 | 1535 | -0.69% | 1508 | 1541 | 2.19% |
| LA14 | 20*5 | 1602.5 | 1637 | 2.15% | 1544 | 1593 | 3.17% |
| LA15 | 20*5 | 1621.8 | 1674 | 3.22% | 1565 | 1597 | 2.04% |
| LA16 | 10*10 | 1148.6 | 1148 | -0.05% | 1148 | 1148 | 0.00% |
| LA17 | 10*10 | 968 | 968 | 0.00% | 968 | 968 | 0.00% |
| LA18 | 10*10 | 1082.4 | 1077 | -0.50% | 1077 | 1077 | 0.00% |
| LA19 | 10*10 | 1110.5 | 1102 | -0.77% | 1102 | 1102 | 0.00% |
| LA20 | 10*10 | 1118 | 1118 | 0.00% | 1118 | 1118 | 0.00% |
| LA21 | 15*10 | 1556.6 | 1542 | -0.94% | 1483 | 1536 | 3.57% |
| LA22 | 15*10 | 1376.2 | 1427 | 3.69% | 1328 | 1387 | 4.44% |
| LA23 | 15*10 | 1525.2 | 1578 | 3.46% | 1475 | 1568 | 6.31% |
| LA24 | 15*10 | 1482.8 | 1533 | 3.39% | 1402 | 1533 | 9.34% |
| LA25 | 15*10 | 1466.7 | 1464 | -0.18% | 1406 | 1448 | 2.99% |
| LA26 | 20*10 | 1980.8 | 2119 | 6.98% | 1870 | 2005 | 7.22% |
| LA27 | 20*10 | 2064.8 | 2148 | 4.03% | 1933 | 2170 | 12.26% |
| LA28 | 20*10 | 2016.7 | 2127 | 5.47% | 1937 | 2168 | 11.93% |
| LA29 | 20*10 | 1898.3 | 1934 | 1.88% | 1764 | 1909 | 8.22% |
| LA30 | 20*10 | 2024.6 | 2048 | 1.16% | 1939 | 2048 | 5.62% |
| LA31 | 30*10 | 2842.8 | 3347 | 17.74% | 2714 | 3013 | 11.02% |
| LA32 | 30*10 | 3106.6 | 3626 | 16.72% | 2928 | 3373 | 15.20% |
| LA33 | 30*10 | 2843.1 | 3009 | 5.84% | 2717 | 3009 | 10.75% |
| LA34 | 30*10 | 2905.7 | 3072 | 5.72% | 2769 | 3072 | 10.94% |
| LA35 | 30*10 | 2938.4 | 3436 | 16.93% | 2757 | 2992 | 8.52% |
| LA36 | 15*15 | 1804.3 | 1658 | -8.11% | 1683 | 1658 | -1.49% |
| LA37 | 15*15 | 1929.4 | 1936 | 0.34% | 1856 | 1913 | 3.07% |
| LA38 | 15*15 | 1734.5 | 1788 | 3.08% | 1665 | 1704 | 2.34% |
| LA39 | 15*15 | 1792.2 | 1792 | -0.01% | 1720 | 1792 | 4.19% |
| LA40 | 15*15 | 1785.4 | 1849 | 3.56% | 1712 | 1771 | 3.45% |

Table 5: Comparison between Tabu Search and S&S on TA1-50 instances

| Instance | Size | 60s | | | 600s | | |
|---|---|---|---|---|---|---|---|
| | | Tabu Obj | S&S Obj | Gap (%) | Tabu Obj | S&S Obj | Gap (%) |
| TA01 | 15*15 | 1769.4 | 1782 | 0.71% | 1761.2 | 1745 | -0.92% |
| TA02 | 15*15 | 1713.6 | 1808 | 5.51% | 1700.0 | 1769 | 4.06% |
| TA03 | 15*15 | 1750.4 | 1786 | 2.03% | 1715.0 | 1675 | -2.33% |
| TA04 | 15*15 | 1682.6 | 1855 | 10.25% | 1659.5 | 1696 | 2.20% |
| TA05 | 15*15 | 1729.0 | 1770 | 2.37% | 1712.2 | 1696 | -0.95% |
| TA06 | 15*15 | 1754.4 | 1829 | 4.25% | 1727.8 | 1806 | 4.53% |
| TA07 | 15*15 | 1780.6 | 1774 | -0.37% | 1753.6 | 1797 | 2.47% |
| TA08 | 15*15 | 1756.6 | 1830 | 4.18% | 1723.8 | 1762 | 2.22% |
| TA09 | 15*15 | 1814.2 | 1882 | 3.74% | 1797.0 | 1882 | 4.73% |
| TA10 | 15*15 | 1762.4 | 1853 | 5.14% | 1728.6 | 1853 | 7.20% |
| TA11 | 20*15 | 2127.6 | 2164 | 1.71% | 2078.4 | 2164 | 4.12% |
| TA12 | 20*15 | 2276.6 | 2536 | 11.39% | 2228.8 | 2235 | 0.28% |
| TA13 | 20*15 | 2134.8 | 2291 | 7.32% | 2099.0 | 2165 | 3.14% |
| TA14 | 20*15 | 2140.2 | 2330 | 8.87% | 2098.4 | 2227 | 6.13% |
| TA15 | 20*15 | 2152.8 | 2501 | 16.17% | 2106.6 | 2199 | 4.39% |
| TA16 | 20*15 | 2236.0 | 2390 | 6.89% | 2225.6 | 2236 | 0.47% |
| TA17 | 20*15 | 2296.0 | 2761 | 20.25% | 2261.8 | 2296 | 1.51% |
| TA18 | 20*15 | 2215.8 | 2444 | 10.30% | 2157.0 | 2387 | 10.66% |
| TA19 | 20*15 | 2190.6 | 2323 | 6.04% | 2129.4 | 2316 | 8.76% |
| TA20 | 20*15 | 2238.6 | 2382 | 6.41% | 2167.8 | 2279 | 5.13% |
| TA21 | 20*20 | 2637.0 | 2770 | 5.04% | 2517.0 | 2770 | 10.05% |
| TA22 | 20*20 | 2536.2 | 2583 | 1.85% | 2437.0 | 2583 | 5.99% |
| TA23 | 20*20 | 2492.6 | 2771 | 11.17% | 2396.8 | 2748 | 14.65% |
| TA24 | 20*20 | 2545.0 | 2829 | 11.16% | 2484.2 | 2829 | 13.88% |
| TA25 | 20*20 | 2487.8 | 2627 | 5.60% | 2394.4 | 2575 | 7.54% |
| TA26 | 20*20 | 2637.2 | 2784 | 5.57% | 2544.6 | 2784 | 9.41% |
| TA27 | 20*20 | 2667.2 | 2721 | 2.02% | 2577.4 | 2721 | 5.57% |
| TA28 | 20*20 | 2545.4 | 2841 | 11.61% | 2471.6 | 2835 | 14.70% |
| TA29 | 20*20 | 2615.2 | 2744 | 4.93% | 2537.6 | 2744 | 8.13% |
| TA30 | 20*20 | 2540.8 | 2702 | 6.34% | 2465.8 | 2678 | 8.61% |
| TA31 | 30*15 | 3358.8 | 4124 | 22.78% | 3189.0 | 3453 | 8.28% |
| TA32 | 30*15 | 3395.4 | 4336 | 27.70% | 3249.4 | 3621 | 11.44% |
| TA33 | 30*15 | 3501.6 | 3980 | 13.66% | 3362.6 | 3595 | 6.91% |
| TA34 | 30*15 | 3474.8 | 3832 | 10.28% | 3285.2 | 3832 | 16.64% |
| TA35 | 30*15 | 3334.2 | 3575 | 7.22% | 3160.6 | 3575 | 13.11% |
| TA36 | 30*15 | 3387.8 | 3691 | 8.95% | 3270.6 | 3647 | 11.51% |
| TA37 | 30*15 | 3478.2 | 3968 | 14.08% | 3324.8 | 3652 | 9.84% |
| TA38 | 30*15 | 3263.2 | 4406 | 35.02% | 3121.4 | 3534 | 13.22% |
| TA39 | 30*15 | 3159.0 | 3309 | 4.75% | 3036.2 | 3309 | 8.98% |
| TA40 | 30*15 | 3270.0 | 3535 | 8.10% | 3117.4 | 3535 | 13.40% |
| TA41 | 30*20 | 3890.4 | 4036 | 3.74% | 3638.2 | 4036 | 10.93% |
| TA42 | 30*20 | 3745.6 | 3979 | 6.23% | 3535.8 | 3979 | 12.53% |
| TA43 | 30*20 | 3618.8 | 3672 | 1.47% | 3460.0 | 3672 | 6.13% |
| TA44 | 30*20 | 3805.0 | 3967 | 4.26% | 3593.0 | 3967 | 10.41% |
| TA45 | 30*20 | 3888.2 | 4067 | 4.60% | 3578.6 | 4067 | 13.65% |
| TA46 | 30*20 | 3867.8 | 4240 | 9.62% | 3610.2 | 4240 | 17.45% |
| TA47 | 30*20 | 3776.0 | 4239 | 12.26% | 3531.0 | 4239 | 20.05% |
| TA48 | 30*20 | 3773.6 | 3977 | 5.39% | 3513.4 | 3977 | 13.20% |
| TA49 | 30*20 | 3694.2 | 4072 | 10.23% | 3480.8 | 4072 | 16.98% |
| TA50 | 30*20 | 3834.2 | 4067 | 6.07% | 3617.6 | 4067 | 12.42% |

Table 6: Comparison between Tabu Search and S&S on TA51-80 instances

| Instance | Size | 60s | | | 600s | | |
|---|---|---|---|---|---|---|---|
| | | Tabu Obj | S&S Obj | Gap (%) | Tabu Obj | S&S Obj | Gap (%) |
| TA51 | 50*15 | 5689.4 | 5904 | 3.77% | 5213.8 | 5904 | 13.24% |
| TA52 | 50*15 | 5703.8 | 5794 | 1.58% | 5228.4 | 5794 | 10.82% |
| TA53 | 50*15 | 5515.4 | 5546 | 0.55% | 5113.6 | 5546 | 8.46% |
| TA54 | 50*15 | 5540.4 | 5809 | 4.85% | 5157.4 | 5809 | 12.63% |
| TA55 | 50*15 | 5577.4 | 5765 | 3.36% | 5080.4 | 5765 | 13.48% |
| TA56 | 50*15 | 5666.0 | 5898 | 4.09% | 5233.6 | 5898 | 12.69% |
| TA57 | 50*15 | 5731.6 | 5816 | 1.47% | 5301.4 | 5833 | 10.03% |
| TA58 | 50*15 | 5833.0 | 6076 | 4.17% | 5397.8 | 6076 | 12.56% |
| TA59 | 50*15 | 5488.4 | 5650 | 2.94% | 5108.6 | 5650 | 10.60% |
| TA60 | 50*15 | 5757.0 | 5814 | 0.99% | 5198.0 | 5757 | 10.75% |
| TA61 | 50*20 | 6542.6 | 6774 | 3.54% | 5198.2 | 6774 | 30.31% |
| TA62 | 50*20 | 6788.8 | 6813 | 0.36% | 6021.4 | 6813 | 13.15% |
| TA63 | 50*20 | 6441.4 | 6294 | -2.29% | 5646.0 | 6294 | 11.48% |
| TA64 | 50*20 | 6320.6 | 6548 | 3.60% | 5576.4 | 6548 | 17.42% |
| TA65 | 50*20 | 6512.0 | 6416 | -1.47% | 5675.2 | 6416 | 13.05% |
| TA66 | 50*20 | 6519.6 | 6738 | 3.35% | 5816.4 | 6738 | 15.84% |
| TA67 | 50*20 | 6567.6 | 6276 | -4.44% | 5745.4 | 6276 | 9.24% |
| TA68 | 50*20 | 6356.4 | 6194 | -2.55% | 5804.2 | 6194 | 6.72% |
| TA69 | 50*20 | 6699.6 | 6580 | -1.79% | 5907.0 | 6580 | 11.39% |
| TA70 | 50*20 | 6764.0 | 6422 | -5.06% | 5882.6 | 6422 | 9.17% |
| TA71 | 100*20 | 17426.6 | 14895 | -14.53% | 12369.4 | 12285 | -0.68% |
| TA72 | 100*20 | 16225.8 | 15763 | -2.85% | 11745.6 | 12534 | 6.71% |
| TA73 | 100*20 | 17370.4 | 15313 | -11.84% | 12078.6 | 12358 | 2.31% |
| TA74 | 100*20 | 16963.9 | 14788 | -12.83% | 12044.8 | 13067 | 8.49% |
| TA75 | 100*20 | 17127.6 | 15151 | -11.54% | 11911.4 | 12156 | 2.05% |
| TA76 | 100*20 | 16578.0 | 14774 | -10.88% | 12223.8 | 12321 | 0.80% |
| TA77 | 100*20 | 17674.8 | 16365 | -7.41% | 12412.2 | 12511 | 0.80% |
| TA78 | 100*20 | 17007.8 | 15014 | -11.72% | 11898.6 | 12807 | 7.63% |
| TA79 | 100*20 | 17145.8 | 15834 | -7.65% | 12118.4 | 12250 | 1.09% |
| TA80 | 100*20 | 16186.4 | 15100 | -6.71% | 11729.0 | 11825 | 0.82% |