# OpenReview forum: "Select and Schedule: An Efficient Hierarchical Optimizer for Blocking Job Shop Scheduling Problem with Massive Jobs"
_ICLR.cc/2026/Conference — ICLR 2026 Conference Withdrawn Submission_

### Official Review · Reviewer_tPXm · 2025-10-24

**Soundness:** 2
**Presentation:** 3
**Contribution:** 1
**Rating:** 2
**Confidence:** 4

**Summary:**

This paper studies the Blocking Job Shop Scheduling Problem (BJSP), which features a no-buffer constraint between machines where jobs must remain on their current machine if the next machine is unavailable. The authors propose a hierarchical Select-and-Schedule (S&S) framework that first selects a small subset of jobs using a learned model, then schedules them using a constraint-programming (CP) solver. Specifically, a Set Transformer is trained via supervised learning to predict machine utilization, which guides job subset selection. Experiments on LA/TA benchmarks report improvements over baselines.

**Strengths:**

1. The paper is well-organized and easy to follow.
1. The S&S method combines the strengths of learning-based methods and traditional CP solvers, which is a reasonable design choice.
1. The S&S method improves performance for BJSP and has potential real-world impact in manufacturing and production scheduling.
1. The learning components achieve near-linear runtime scaling versus the number of jobs and competitive quality at longer budgets.

**Weaknesses:**

1. The paper studies a specific problem variant that may have a limited impact on the broader ICLR community. More critically, the conceptual novelty is insufficient. The hierarchical framework of using learning to decompose problems followed by optimization has been extensively explored in neural combinatorial optimization, including scheduling, vehicle routing, graph CO, etc. The paper closely resembles existing work (Chen et al. 2024, Li et al. 2025, Lee et al. 2019) in terms of framework design, heuristic design, and network architecture, without sufficiently differentiating its unique contributions.
1. The evaluation considers inadequate baselines. 1) For classical solvers, the paper only includes Tabu search and the CP solver. More state-of-the-art classical heuristics should be adapted and compared. 2) For learning-based baselines, the paper does not compare with any of them. There are lots of DRL-based methods or learning-guided solvers that have been developed for JSP and can be adapted for this BJSP problem with minimal changes. 3) The paper claims that block constraints "substantially increase problem complexity, which in turn limits most existing scheduling algorithms to small-scale instances," but provides no analysis or supporting evidence for this claim.
1. The experimental analysis lacks depth across multiple dimensions. Tables 1 and 2 do not report detailed runtime analysis, making it difficult to assess the computational efficiency. The ablation studies are severely limited. There is only one ablation that removes the entire learning component, but more meaningful comparisons should include well-designed heuristics for job selection beyond the random S&S baseline to truly demonstrate the necessity of learning. Component-wise ablation studies of the network design choices shown in Figure 2 are entirely missing. Additionally, the paper lacks any generalization analysis showing how the method performs across different problem distributions or instance characteristics, and there is no evaluation of robustness under more complex scenarios or varying distributions.
1. Even with limited evaluation, the effectiveness of the proposed method still seems questionable. The method shows mixed performance compared to Tabu search at shorter runtimes (60s) and exhibits significantly larger gaps at longer runtimes (1800s), suggesting the method may be trapped in local optima and that the neural network may have poor generalization for longer planning horizons. Furthermore, Table 3 reveals that compared to random S&S, the learning provides only marginal improvements, often around 1%. This raises questions about whether simple heuristics could achieve similar performance and whether the learning component is truly necessary, as I also mentioned in point 3.

**Questions:**

Please see my weakness.

---

### Official Review · Reviewer_R7CC · 2025-10-29

**Soundness:** 3
**Presentation:** 3
**Contribution:** 2
**Rating:** 4
**Confidence:** 4

**Summary:**

This paper studies the Blocking Job Shop Scheduling Problem (BJSP), which requires a job to remain on its current machine until the next machine is available. Due to the nature of the blocking constraint, it restricts the number of jobs processed concurrently and hence reduces the number of candidate jobs. Due to this property, this paper learns to select from this small subset of candidate jobs and defers the scheduling of these jobs with a solver. Empirically, they show their learning method is able to scale much better when handling larger instances.

**Strengths:**

1. It makes a lot of sense to leverage the blocking constraint to reduce the complexity of learning.

2. Results on large scale BJSP seem very competitive under a short time limit.

**Weaknesses:**

1. The premise of this paper relies on the blocking constraint in BJSP, but BJSP is just a very specific type of scheduling problem, and it seems challenging to generalize the idea of this paper to other scheduling variants.

2. While the idea makes sense, it seems a bit too natural to me under the BJSP setting, so I’m concerned about the novelty of this paper.

3. Only two small sets of instances were tested (TA, LA). Can the authors test on more realistic test benchmarks?

**Questions:**

1. Table 1, seems like under a larger time limit, S&S is not as competitive as Tabu search. Is it because the S&S more easily plateaus in a local optima?

2. Table 3, comparing S&S and random S&S, the learned version S&S seems to have a large compute overhead. Can the authors compare S&S and random S&S over the same solve time?

3. Small comment: can the authors bold the best results for the ease of reading?

---

### Official Review · Reviewer_JFk7 · 2025-11-01

**Soundness:** 1
**Presentation:** 2
**Contribution:** 2
**Rating:** 2
**Confidence:** 4

**Summary:**

The paper introduces Select and Schedule (S&S), a hierarchical optimization framework designed to efficiently solve large-scale instances of the Blocking Job Shop Scheduling Problem (BJSP). The core insight of the paper is that the blocking constraint, typically viewed as a source of difficulty, also naturally restricts the number of jobs that can be processed concurrently (bounded by the number of machines). Building on this idea, S&S splits the instances into subproblem. They train a network that predicts the utilization of a given set of jobs. Based on this network, subsets of jobs are selected. A CP solver is then used to generate a schedule for the subset. This selection and solving procedure run iteratively until the full schedule is created. The experiments demonstrate that the approach improves over the state-of-the-art tabu search method for short runtime scenarios (60 s) and that large-scale instances can be solved relatively quickly due to the near linear computational complexity.

**Strengths:**

1.	The proposed method is based on relevant and logical insights of the problem, and this approach has not been implemented before for the blocking job shop scheduling problem.

2.	The approach is explicitly linked to the theory of submodular functions, providing an approximate theoretical basis for the performance.

3.	The proposed approach shows good computational complexity and performs well on large instances, beating the state-of-the-art approach in scenarios with short available runtimes and approaching its performance with longer runtimes.

**Weaknesses:**

1.	The proposed methodology has several issues and some questionable design choices.

- In the neural network, the three input features of each job are embedded separately. Usually, all features are passed through a layer together. There is no apparent reason the work deviates from this.

- The machine ID is used as a numerical input feature. However, this should be one-hot encoded. The difference between machines 1 and 2 or machines 1 and 7 are semantically the same, but if you represent it in a numerical way, this is not the case.

- The neural network can only handle instances where all jobs have an equal number of operations (and this number is equal to the number of machines). This is a severe practical limitation. In NCO literature many (graph-based) networks can handle any combination of jobs, operations, and machines.

2.	The experiments are limited and not very convincing. Some issues are:

-	No experiments are shown on to evaluate the value estimation neural network training procedure. The paper does not show the loss or any other evaluation metric to evaluate how well the network approximates utilization. Also, the paper does not compare the proposed architecture with other networks and misses an ablation study on the design components. Questions about this performance are raised further by the relatively small performance difference between random S&S and S&S.

-	The comparison with different approaches is limited. For the Taillard and Lawrence instances, the approach is only compared with Tabu Search. It should at least compare on these instances with the random S&S approach to show that the learning component is also valuable in these instances. And ideally it should also include the comparison with the CP solver here. Simultaneously, the Tabu Search is not used as a baseline for the other synthetic instances, for which no reason is given.

-	While the paper states that the approach can be used for both the BNS and BWS scenario, the paper does not show experiments for this. The value of the approach would be much more emphasized if promising results on both scenarios are shown.

3.	The framing of the work in existing literature is lacking.

-	The related works section only includes some exact and metaheuristic methods proposed for the BJSP. However, there have been many works presented on machine scheduling using learning approaches. There have been many neural combinatorial optimization (NCO) works on the JSP and FJSP, proposing various network architectures and training mechanisms for solving such problems directly. Moreover, hybrid machine learning/optimization methods have also been proposed for various combinatorial optimization problems. The most relevant work by Li et al. (2025) should be elaborately discussed, since it offers a machine learning/constraint programming combination used to address large scale scheduling problems. The paper should include a clear description on why this approach is not applicable here. And if it is applicable, a performance comparison should be made.

-	The value of the approach seems quite limited to this specific blocking job shop scheduling problem and in many cases, it does not outperform the state-of-the-art approach. The paper should discuss more elaborately in which cases the approach is applicable. Is it also applicable to other scheduling problems with different objectives?

4.	The writing in the paper is of insufficient quality.

-	There are several issues in the mathematical definitions and notations.

-	Multiple important methodological details are not explained clearly.

-	Multiple experimental details are not explained appropriately.

-	Several descriptions and sentences are ambiguous.


Textual mistakes:

-	Line 36: “which frequently” should be “which is frequently” or “frequently”.

-	Line 47: “such circular dependencies”. The paper did not yet introduce any such dependencies.

-	Line 137: The cited work by Luo et al. (2021) uses a DRL approach to tackle a dynamic partial-no-wait multi-objective FJSP. This is very different from what the paper argues that it does.

-	Line 151: m_{j,k} \in \mathcal{M} This notation is not good. It makes it seems like the set \mathcal{M} of machines consists of all machines where each operation j,i has a specific machine. It should be more clearly define that this is the set of machines and multiple operations can use the same machine.

-	Line 151-152: Either the processing times should be in \mathbb{N} or the start times should be in > 0. Now it is inconsistent.

-	Line 152: Missing space after completion time sentence.

-	Line 153-154: JSP and BJSP have been introduced before already.

-	Equation 1: For understanding of people less familiar with these functions it would be good to add a sentence that explains this property (i.e., the marginal gain of adding an element diminishes with for larger sets.). It is not strictly necessary but would be beneficial for some readers.

-	Line 159: the paper used N here, but in the remainder of the text uses \mathcal{N}.

-	Line 164: The symbol S is overloaded for subsets of jobs and the set of start times in a schedule.

-	Line 258: “all operations in \sigma”. Should “operations” be “jobs”? After an operation completion, your set of in-process jobs does not change. Looking at Figure 2, it also seems to indicate jobs. This is quite a crucial difference.

-	Line 274: Use vector notation for x.

-	Line 281: o_{ijk} should be o_{jk}.

-	Line 286: missing space between Blocks and (SAB).

-	Equation 8: use of J^’ is ambiguous. Better to refer to the block using e.g., J^(l+1).

-	Line 303: What is W? A matrix of what shape?

-	The paper uses multiple different notations for instance sizes. It introduces that they use |J|x|M| but then continues to also use (J, M) and J*M later in the paper.

-	Line 360: What is the “baseline”?

-	Line 361: What is “the unlimited setting”?

-	Add an explanation of how the gap is computed.

-	Line 703: “cases(t)” what is this t?

-	Line 720: “almost 100%” what is the exact number?

**Questions:**

Main suggestions:
The authors should address the concerns and suggestions raised in the weaknesses. First, the methodological shortcomings should be addressed. Either very convincing arguments should be made for some of the design choices, or the methodology should be improved. Second, the experimental section should be considerably expanded. Third, the related works sections, as well as the framing of the paper related to other works should be expanded. A discussion of the broader potential of the approach to more problems than just this BJSP variant should be discussed to show its relevance to the broader community. Fourth, the quality of writing, mathematical preciseness, and the explanation of the methodology and experimental setup should be improved.

Below several smaller questions and suggestions are discussed. These are partially overlapping and partially in addition to the above improvement directions.


- To what extent will the approach work for other objectives, like tardiness, costs, et cetera. Can you discuss how you expect it to work and what modifications should be necessary?

- In section 4 it is left unclear how the inference procedure works. The paper should more clearly explain the exact approach. When exactly do you use GMCC as opposed to greedy insertion? How do you do the greedy insertion? Does it happen that the number of jobs in the subset is smaller than the number of machines because the value network gives a smaller value for this larger set? Or is it always full? Clearly explain each step. How do you select the initial set and is it correct that after that there is always one removal with one replacement? Be precise.

- There exist many graph neural networks for job shop scheduling problems that can handle changing numbers of operations, machines, and jobs. Why did the authors not use any of these approaches as an alternative for the less flexible chosen set transformer network? They should clearly explain why and/or add an experimental justification for this.

- In Equation 5, what is the difference between Linear and Embed?

- Equation 6, what is meant by flattening? Maybe concatenating?

- Equation 9, why do you use pooling by multi-head attention? Did you compare with simple pooling methods?

- The mathematical description of the pooling by multi-head attention should be rewritten. This notation seems inconsistent with the description of PMA from the set transformer paper by Lee et al. (2019). The paper does not include the feedforward and skip-connection as well as the layer normalization. Also, the paper is not correctly describing the multi-head attention. I suggest rewriting this part and more closely follow the multi-head attention block structure from the paper by Lee et al. (2019) (which you already use mostly in equations 7 and 8). Also, what number s do you use? Is one not sufficient in this case?

- To what extent is supervised learning appropriate here compared to e.g., reinforcement learning? Discuss the implications, advantages, and disadvantages of using supervised learning here?

- You should explain in more detail what the training instances for the supervised learning approach look like and how the training procedure works. How large are the instances, with how many jobs, how many machines, how many operations? Since supervised learning is used, these training instances should be as similar as possible to what you encounter during inference. We cannot judge if this is appropriately handled and the methodology is appropriate. Is one network trained per instance size or multiple different ones? What is the effect of the training instances on the performance and how robust is the network? Also, what is the time limit for the CP solver in training?

- Line 325: What is the "second schedule-preserving strategy"?

- Which constraint programming solver does the paper use?

- Can you clarify what is meant by “randomly but with the same selection procedure” for R-S&S?

---

### Official Review · Reviewer_5Cyy · 2025-11-03

**Soundness:** 2
**Presentation:** 3
**Contribution:** 2
**Rating:** 4
**Confidence:** 5

**Summary:**

This paper addresses the significant scalability challenges of the Blocking Job Shop Scheduling Problem (BJSP) by introducing a novel hierarchical optimization framework. The authors' key insight is that the blocking constraint, while a primary source of complexity, naturally limits the number of jobs processed concurrently. They leverage this by decomposing the problem: a higher-level neural network selects a small, critical subset of jobs from the candidate pool, which a lower-level solver then schedules efficiently. This two-stage approach reduces computational complexity dramatically, enabling the method to scale almost linearly with the number of jobs. Experimental results demonstrate that this framework not only achieves an average 11% improvement in solution quality on large-scale benchmarks but also efficiently handles previously intractable larger instances within reasonable runtimes.

**Strengths:**

1. The target and motivation of this paper are clearly stated, which helps the audience (even with limited background knowledge) understand the context and rationale of the proposed method smoothly.

2. The method is well motivated.

3. The performance is promising compared to Tabu

**Weaknesses:**

1. The idea of assign-and-solve is not new in learning for solving COPs, for example: https://arxiv.org/abs/2209.06094

**Questions:**

1. Briefly mention the related work in the broader area of learning for manufacturing scheduling would be more beneficial to the audience's interests. The current version only mentions works from the traditional scheduling domain, which makes the whole background more OR-oriented.

2. What different the performance would be if we use different solvers? I would like to see this performance.

3. Similar to 2, can we use a non-learning method for job selection? If yes, how the performance compared with the learning based one?

---

### Note · Authors · 2025-11-12

**Comment:**

We sincerely thank all the reviewers for taking the time to review our paper. The reviewers provided quite a few constructive comments, which indeed highlighted areas we had previously overlooked in our work. We will carefully consider their insightful suggestions to improve our paper. Once again, we thank all the reviewers for their efforts.

**Withdrawal Confirmation:**

I have read and agree with the venue's withdrawal policy on behalf of myself and my co-authors.